# Gut/Oral Bacteria Variability May Explain the High Efficacy of Green Tea in Rodent Tumor Inhibition and Its Absence in Humans

**DOI:** 10.3390/molecules25204753

**Published:** 2020-10-16

**Authors:** Guy R. Adami, Christy Tangney, Joel L. Schwartz, Kim Chi Dang

**Affiliations:** 1Department of Oral Medicine & Diagnostic Sciences, Center for Molecular Biology of Oral Diseases, College of Dentistry, University of Illinois at Chicago, 801 South Paulina Street, Chicago, IL 60612, USA; joschwar@uic.edu (J.L.S.); kimchidang15@gmail.com (K.C.D.); 2Department of Clinical Nutrition, College of Health Sciences, Rush University Medical Center, 600 South Paulina St, Room 716 AAC, Chicago, IL 60612, USA; Christy_Tangney@rush.edu

**Keywords:** catechin, gene expression, mucosa, oral squamous cell carcinoma

## Abstract

Consumption of green tea (GT) and GT polyphenols has prevented a range of cancers in rodents but has had mixed results in humans. Human subjects who drank GT for weeks showed changes in oral microbiome. However, GT-induced changes in RNA in oral epithelium were subject-specific, suggesting GT-induced changes of the oral epithelium occurred but differed across individuals. In contrast, studies in rodents consuming GT polyphenols revealed obvious changes in epithelial gene expression. GT polyphenols are poorly absorbed by digestive tract epithelium. Their metabolism by gut/oral microbial enzymes occurs and can alter absorption and function of these molecules and thus their bioactivity. This might explain the overall lack of consistency in oral epithelium RNA expression changes seen in human subjects who consumed GT. Each human has different gut/oral microbiomes, so they may have different levels of polyphenol-metabolizing bacteria. We speculate the similar gut/oral microbiomes in, for example, mice housed together are responsible for the minimal variance observed in tissue GT responses within a study. The consistency of the tissue response to GT within a rodent study eases the selection of a dose level that affects tumor rates. This leads to the theory that determination of optimal GT doses in a human requires knowledge about the gut/oral microbiome in that human.

## 1. Green Tea Catechins

Green tea (GT) derived from the leaves of the Camellia sinensis plant is a rich source of the polyphenols known as catechins. A 240 mL or 8 ounce serving of GT contains, in solution, 300 mg catechins: (-)- epigallocatechin-3-gallate (EGCG), (-)-epigallocatechin, (EGC), (-)-epicatechin-3-gallate (ECG), (-)-epicatechin (EC) [1], and approximately 30 mg of the stimulant caffeine. The catechins are potent antioxidants that can react with and reduce many different reactive oxygen species [2]. While once thought to inhibit carcinogenesis chiefly by inactivation of dietary oxidants, catechins have been shown to have additional properties inside cells that may contribute to the perceived health benefits of drinking GT [3,4]. These include interactions with intracellular proteins so to alter: apoptosis, transformed cell proliferation, angiogenesis, DNA repair, and enzymatic detoxification of ROS, etc. [2,4,5]. This review will seek to shed light on why GT or GT catechins are verified oral cancer preventatives in animal models while there is much less evidence for this in humans [4,6].

## 2. GT Inhibition of Rodent Cancer

Published reviews detail the many rodent studies documenting the ability of GT extract or GT polyphenol consumption to prevent digestive tract tumors [6,7]. Studies of the oral cavity and the esophagus include usage of hamster, rat, and mouse models to show that GT extract or purified polyphenols in drinking water can inhibit the induction of tumors by various carcinogens at both sites [8,9,10,11,12,13]. Additional studies show efficacy of GT in preventing cancer of the colon and liver, and non-digestive tract sites such as prostate and lung, and that this inhibition can occur whether the GT or GT polyphenols are present during or after carcinogen exposure [2,14,15]. There are notable exceptions where no inhibition of tumor formation was observed, even in cases where genetically identical mice were used [6,16,17,18,19]. However, at least five published studies alone have shown a cancer preventive effect of GT or GT polyphenols on oral cancer induced by 3 different carcinogens [8,9,12,20,21]. GT form or method of application may have differed, and dose may have had some species specificity, but all these studies saw a positive result as shown in Table 1.

## 3. GT Inhibition of Human Cancer

Human studies have not shown the same consistently high level of efficacy of GT or GT polyphenols in prevention of oral cancer or any other cancers [4,6,7]. Epidemiological studies of esophageal cancers revealed overall little or no association between GT drinking and cancer rates [20,21]. Interestingly, in cohort analyses stratified by sex, protective associations between GT and esophageal cancer were observed for Chinese women [21], and in a case-control study for non-tobacco/non-alcohol users and women users [20]. Studies of oral cancer are similarly variable with limited evidence for GT drinking being a cancer preventive based on epidemiology, though curiously there was a tendency for a benefit in females in a prospective cohort study [22]. An early randomized trial of oral squamous cell carcinoma (OSCC) prevention showed reduction in dysplastic lesions by consuming a GT extract in capsules combined with direct application of 1 g GT extract to the lesion [23] but a later trial showed no statistically significant benefit of GT in capsules as shown in Table 2 [24]. Notably, GT polyphenols are typically consumed in capsules when tested in recent clinical trials on cancer prevention, unlike human epidemiological studies, which may contribute to results [24,25,26]. Conflicting findings have also been seen for GT consumption and incidence rates for a number of other cancers, with modestly lower rates of liver and prostate cancer of self-reported tea drinkers based on meta-analysis [27,28,29,30]. Furthermore, recent trials designed to examine GT effects on breast cancer risk noted liver toxicity among 5% of the study subjects taking capsules with the equivalent of 5 cups decaffeinated GT daily [26]. Overall, a clear association between GT or GT polyphenol consumption and human cancer prevention has not been verified.

## 4. Significance of Studying GT Effects In Vivo

A lack of understanding of GT effects on cells relevant to cancer inhibition has made assessment of the effects of GT on cancer prevention more difficult. Some years ago, it was suggested that GT catechins do not work in vivo chiefly as antioxidants that neutralize oxidizing molecules, but may instead work as regulators of cell function often inside cells [14,31]. While much is known about GT catechin effects on cell lines in vitro [27]), forms of GT catechins in vivo may be quite different than ones seen in vitro, making it difficult to know what in vitro findings are relevant in vivo [32,33,34]. Plasma and cellular levels of intact catechins are often quite low due to poor uptake by gut epithelium, compared to concentrations used in in vitro studies, as commented on by Yang and Wang [4]. The major tea catechins including EGCG, as predicted by Lipinski’s rule of five, due to large size, presence of hydrated shells, and their polarity, are poorly absorbed by cells [4,13,35,36,37]. In the case of oral epithelium, which is exposed to undiluted GT in tea drinkers, concentrations of catechins are high, hundreds of micrograms per mL, but exposure is transient, so absorption of the unaltered molecules into cells is suboptimal. Thus, to model catechin effects in human tea drinkers, it may be best to focus on in vivo GT studies. On a positive note, there has been some progress in determining GT-induced changes in global gene expression in vivo in rodents, providing information that may inform us on the mechanism of cancer inhibition.

### GT and Gene Expression

Assaying RNA or protein level changes in tissue after consumption of potential bioactive compounds, such as GT, is a rapid method to show if the compound has an effect on the tissue, and may help discern if effects relevant to cancer inhibition occur. There are a limited number of studies published on GT polyphenol effects on epithelial gene expression (or RNA levels) under conditions of carcinogenesis in vivo in rodents but they show clear effects. A lung cancer induction model in mice has been used to identify gene expression changes related to inflammation and regulation of cell proliferation that occur weeks after EGCG exposure in lung adenoma tissue [38]. Analysis of miRNA expression, a major regulator of gene expression in cells, in the same tissue showed changes in miRNAs with EGCG exposure in the same early tumors [39]. These changes parallel the decrease in progression of adenomas to carcinoma due to EGCG in that tumor model and argue that GT exposure is responsible for changes in epithelial gene expression. Multiple studies have shown that with GT exposure or EGCG consumption, tumor-relevant changes in gene expression occur in various rodent tissues [40,41,42]. There is also extensive evidence of changes in gene expression on the protein level in colons of rodents prone to this cancer that drink or eat GT catechins in their water or food [43,44,45,46]. Similar findings have been shown in rodent models of other cancers as reviewed [4].

Published studies of epithelial gene expression changes, induced by catechin or GT extract consumption, in humans, are rare. In a randomized placebo-controlled trial of subjects with oral premalignant changes, immunohistological examination of oral mucosa after 3 months of GT-extract consumption revealed no changes in a range of proteins after exposure. In a subset of those with reduced dysplasia, Cyclin D1 and Vascular Endothelial Growth Factor (VEGF) mucosal levels decreased [24]. Brush biopsy offers a noninvasive and validated method optimized for miRNA measurement of oral epithelial cells [47]. After 4 weeks of GT drinking, human tissue exposed to probably the highest concentration of undiluted tea in the body, the tongue epithelium, showed on average no changes in gene expression due to inter-subject variability in levels of miRNA. Only after differential co-expression analysis, which can correct for a lack of a response in some subjects, did GT-induced changes in miRNA expression become evident [48,49,50,51]. Non-supervised hierarchical clustering of oral epithelial sample miRNA revealed a group of 5 out of 14 subjects who, after consuming GT, differed most from control subjects who did not consume GT [48]. One might conclude either GT had subtle effects on the tongue epithelium or the effects were subject specific. Another study by Choi et al. studied blood samples from human patients who consumed 300 mL GT per day. They saw subtle, less than 50%, changes in levels of two redox-linked cytoprotective enzymes [52] and no change in mRNAs were observed. This one study stands in contrast to the many studies that have shown GT-induced changes in gene expression in rodent blood cells and epithelium [38,39,40,41,42,43,45,46]. Interestingly, human studies of bioactive compounds, such as the catechins and other polyphenols, in general, rarely show changes in levels of specific RNAs. Pokimica et al. have commented on the great variance in gene expression changes with human exposure to GT polyphenols and many other bioactive compounds [53]. Notably, most studies failed to show changes rising to statistical significance in any genes after polyphenol consumption. While technical problems may contribute to low precision in measurement [50], it is also possible different humans are different in their responses to polyphenols like the catechins.

## 5. GT Extract and GT Catechin Interactions with Oral and Gut Bacteria

### 5.1. GT Extract and Catechin Effects on Oral and Gut Bacteria

There is ample evidence that GT catechins are selective antibiotics in vitro, though this effect is concentration-dependent [54,55,56]. Bacteria diminution is thought to occur via membrane damage, prevention of biofilm formation, enzyme poisoning, etc. [56,57,58]. Pioneering studies in mammals that focused on several oral bacteria at a time noted changes in oral microbiota with tea or tea extract consumption with consistent reductions in *clostridium* and increases in *bifidobacterium* and *lactobacillus* (Table 3). Two recent human studies broadly examined oral microbiome at genus and species level, after GT or GT extract consumption [48,59]. The first study examined samples from two oral mucosal sites [59,60]. The second examined saliva, a mixture of microbes shed from all mucosal and tooth surfaces [61]. After two and four weeks of GT consumption, changes in frequencies of multiple taxa were observed throughout the study populations in both studies. A third microbiome study reported on GT polyphenol effects on gut microbiome but only examined changes on the phylum level [61]. In sum, the reproducible effects of GT consumption on oral and gut genera in human and nonhuman studies suggest GT has direct effects on gut/oral bacteria.

### 5.2. Gut and Oral Bacteria May Metabolize GT Polyphenols

There is much evidence that gut bacteria metabolize dietary polyphenols. This can, in theory, transform them into more bioavailable forms and stimulate uptake into digestive tract epithelium [68,69,70,71]. Years ago, it was shown that GT catechins can be metabolized by intestinal bacterial enzymes. A number of groups have characterized these reactions chiefly by two approaches: (1) examination of products present in blood, urine and feces post-catechin consumption; and (2) incubation of catechin of interest, chiefly EGCG with rat gut contents in vitro. Of the 4 major catechins in GT, EGCG, EGC, ECG, and EG, peak blood levels, in humans and rodents, reach about 1.0 μM for each due to low absorption by digestive tract epithelium [4,32,72]. Metabolism by gut microbes of ingested EGCG, EGC, and EC creates 5-(3′4′-dihydroxyphenyl)-γ-valerolactone, 3-(3-hydroxyphenyl) propionic acid and other compounds [73,74,75,76,77]. Braune and Blaut have described the inroads that have been made in identifying the colonic bacteria that perform the hydrolysis of ester bonds, reductive cleavage of the C ring and further reactions of GT catechin catabolism (Figure 1) though little is known about the enzymes responsible [78]. Recent work correlating specific taxa with levels of the first catechin metabolites has hinted at a number of human gut bacteria as candidates to contribute to this process including *R. bromii* and *Eggerthella*, the latter already known to be involved [79]. Studies of humans are beginning to reveal inter-individual variability in GT catechin metabolism by digestive tract bacteria relevant to health [80,81]. In a small study, human subjects who consumed high levels of green tea polyphenol mixture, without caffeine, over 8 weeks, had differently hydroxylated phenyl-γ-valerolactones and 3-(3′-hydroxyphenyl)propionic acid in their urine suggesting different levels of enzymes performing the C-ring cleavage of GT catechins, the dehydroxylation of phenyl-γ-valerolactones and the conversion of t phenyl-γ-valerolactones into 3-(phenyl)propionic acids, [74,77,82,83]. Some GT metabolites are more readily taken up by cells than the parent compounds and still retain biological activities relevant to carcinogenesis at least in vitro [84,85]. More extensive metabolism of GT catechins results in biologically inactive phenols that are excreted. It has been shown that removal of gut bacteria by extended antibiotic exposure results in increased levels of intact GT catechins in feces and serum showing the importance of the microbiota in GT metabolism [86]. Unfortunately, the study did not report on levels of absorbed GT catechin metabolites. The mouth is also a rich site of microbes, and there is some evidence that these microbes can metabolize tea catechins, to more readily absorbable valerolactones, though the process is not well described [87]. Importantly digestive tract bacteria, best documented for the colon, have the ability to metabolize catechins, supplementing the body’s own metabolism of these molecules [32]. Whether the oral epithelium absorbs the bulk of GT catechins and their metabolites directly from the tea/salvia or via blood is not known.

One may speculate that gut/oral microbiota may play a role in the differences in rodent versus human responses to GT. In animal studies of GT, GT catechins or other chemicals have been shown to induce physiological changes, gene expression changes, and prevent and slow progression of a variety of cancers in the animals, which are housed together. Animals in the same cage show high levels of similarity in gut microbiome [88,89,90]. Rodents in a single facility are known to show many more similarities in microbiota than animals in a different facility, due to shared diet and other unknown factors [90,91,92]. In a study with all mice in a single room, the way almost all rodent studies of GT are done, gut microbiota are more similar than not, especially in cases where the animals are also homozygous, as genetic differences can contribute to mucosal microbiota identity [93]. A separate study done at a second facility would likely have different gut/oral microbiota [90,91,92,93]. Since gut and likely oral microbiota have the ability to modify GT catechins, and change bioavailability of tea catechins, one would expect that responses to poorly bioavailable GT compounds would be uniform within a cohort with similar oral microbiota and gut microbiota [68,69,70,71]. This would explain the ease in detecting changes in gene expression brought about by GT exposure in rodent studies, as intraclass variation would be minimal within the study as long as the mice were housed together. A second issue is that the uniformity of GT effects within a study would make it easier to optimize dose to see maximal phenotypic changes and to avoid toxicity. The optimal GT or GT extract dose at a second study site would likely be different but would, for the same reason, be easy to optimize. For example, too high a dose would be toxic to all rodents, not just a few, and would be harder to miss. Usage of an optimal GT dose would make it more likely that GT would be active across all animals and have an effect on tumor rates. In some studies where no optimization of GT dosage for the individual cohort was done, one would predict no effect on tumor rates might be seen and there are a small number of published rodent studies where GT did not affect tumor rates [4,15,16,17,18]. Consistent with this model is that the dosage of GT polyphenols chosen to inhibit OSCC in mice, and not be toxic, is variable (Table 1), as seen with other cancers [8,12,94,95].

## 6. Model for How Variable Gut and Oral Microbiota may Affect GT Studies on Humans

People who are non-cohabitating show a great variety of oral and gut microbiota which is reduced in those who live together [96,97]. Diet [98,99], gut/oral health [100,101], and drugs [102,103] may further influence gut and oral microbiota. Clinical studies on GT-based prevention do not normally account for variation of aerodigestive tract microbiota or the foods, beverages, and medications ingested. As a result, one would predict that responses to supplemental polyphenol and GT itself would be variable. Whether one is assaying changes in cell function [104] or histological changes in cancer-prone sites [24], rarely is a consistent net positive result seen across a sample of human subjects. This may be due to the differences in GT metabolizing digestive tract microbes that we suggest are crucial for catechin uptake, turnover, and/or function [35,70,105]. As a result, levels of GT extract that may be appropriate for most individual humans are much too high, for example, in subjects with gut/oral microbes most efficient at converting GT polyphenols to metabolites that are bioactive. Conversely, a study on a rodent cohort consuming GT polyphenols, with bioactivity dependent on digestive tract microbiota which vary little, would reveal consistent changes in gene expression. This would ease GT dosage optimization in a rodent study and make it fairly likely that changes in cancer incidence would be observed (Figure 2). In clinical trials, using humans with variable gut/oral microbiota, that would not be the case [106]. For example, humans with high levels of gut/oral bacteria that metabolize GT polyphenol to functional, readily absorbed metabolites might show toxicity, while those with gut/oral bacteria that lack this metabolic activity may show no effect [26]. With heterogeneous populations, a dosage would be chosen tolerable to all but the most sensitive subjects. It would vary little between studies and be on-average ineffective and that is what has been observed (Table 2). This is in contrast to rodent studies where levels of catechins given to experimental subjects vary between studies (Table 1). One possible solution is to artificially convert GT polyphenols to forms that are more readily absorbed by cells [35,107]. Another would be to characterize subject specific GT toxicity, possibly by measuring each subject’s gut/oral microbiota and its ability to activate and inactivate GT catechins, prior to entry in the trial, or more directly measuring GT metabolites after the study starts.

This comparison of rodent and human studies points out a role for digestive tract microbiota in the disparate results in inhibition of OSCC and other cancers in rodents versus humans. It is not evident what is more important in the oral cancer process, how GT changes the oral microbiome, or direct effects of GT on the oral mucosa. Other differences in human and rodent studies, such as methods of ingestion, liquid or capsule, and sex of the subjects, may contribute to differences in oral cancer inhibition in human versus rodents.

## Figures and Tables

**Figure 1 molecules-25-04753-f001:**
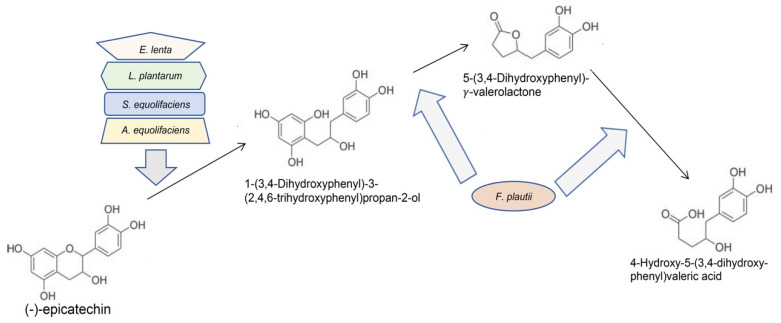
Initial products of metabolism of green tea (GT) (-)-epicatechin (EC). Four gut bacteria have enzymes to execute the first C ring cleavage step, while F plautii has been shown to be capable of further metabolism.

**Figure 2 molecules-25-04753-f002:**
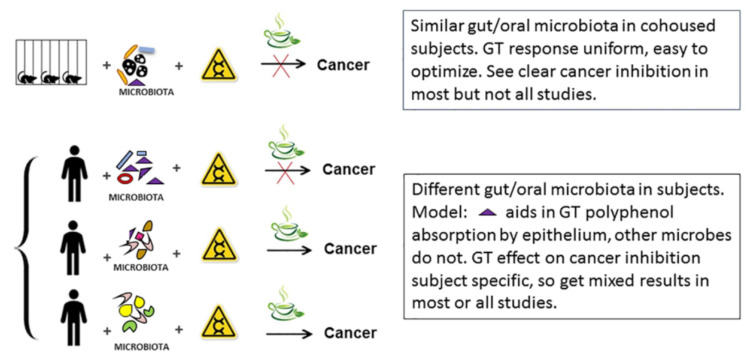
Model of shared oral/gut microbiota on ability of GT and GT polyphenols to prevent cancer. 
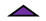
 represents a bacterium that is efficient at converting GT polyphenols to bioactive forms readily absorbed by digestive tract epithelial cells. The human with high levels of this bacterium would readily process GT catechins to active, absorbed forms, and show a response. The other two humans who lack this bacterium would not.

**Table 1 molecules-25-04753-t001:** Extract/polyphenol effects on rodent oral squamous cell carcinoma models.

Species	Catechin Mixture	Delivery of GT	Dose of GT Extract for Equivalence in Polyphenols	Duration of GTE or GTP Exposure	Inhibition of Incidence ^6,7^	Decrease inTumor No.	Decrease in Tumor Vol.	Study
Wistar albino rats, Male ^1^	200 mg/kg GT polyphenol, daily	Drinking water	600 mg/kg GT	12 weeks	NS	44%	58.6%	11
Syrian Golden Hamster, Male ^2^	600 mg/kg GT extract daily	Drinking water	600 mg/kg GT	18 weeks	NS	35.4	57.3	19
Syrian Golden Hamster, Male ^3^	1500 mg/kg GT extract, daily	Drinking water	1500 mg/kg GT	17 weeks	NS	42.1	67.3	9
C3H/HR syngeneic mouse ^4^	25 mg/kg GT polyphenol	IP injection	75 mg/kg GT	21 days	NS	ND	43.6	8
Swiss albino mice, Male ^5^	8 mg/kg GT polyphenol	Oral gavage	0.002 mg/kg GT	24 weeks	100%	ND	ND	12

^1^ GT polyphenol given after 4-Nitroquinoline 1-oxide (4-NQO) oral application completed. ^2^ GT extract given for 18 weeks, after 7,12-dimethylbenz[a]anthracene (DMBA) oral application completed. ^3^ GT extract given 2 weeks before and then concurrent with 15 weeks DMBA oral application. ^4^ ECGC injections after syngeneic mouse tumor cell injections. ^5^ ECGC given daily for last 24 weeks of *N*-Nitrosodiethylamine (NDEA) oral application. ^6^ Inhibition of incidence of OSCC; or for Swiss albino mice only, moderate to severe dysplasia of tongue. ^7^ NS: not significant.

**Table 2 molecules-25-04753-t002:** Green Tea Polyphenol Clinical Trials.

Catechin Mixture	Delivery	Dose of GT Extract for Equivalence in Polyphenols	Equivalent Level GT as Beverage (Approx)	Duration of Exposure	Study
3 g extract/day	Capsule and topical	55 mg/kg daily	Unknown	6 months	[23]
2.6–5.2 g extract/day	Capsule 3/day	41–83 mg/kg daily	3.5–5 cups of 240 mL tea each	12 weeks	[24]
1.3 g catechin/day	Capsule 4/day	60 mg/kg daily	4.5 of 240 mL cups tea	12 months	[26]

**Table 3 molecules-25-04753-t003:** Effects on Gut Bacteria.

Catechin Mixture	Experimental Subjects	Time of Exposure	Lower Concentration Gut Bacteria Post GT or GTE	Higher Concentration Gut Bacteria Post GT or GTE	Study
Green tea extract 0.2% in feed	Pig	2 weeks	*Bacteroidacea, Clostridium perfringens*	*Lactobacillus*	[62]
Polyphenon G ^1^ 0.2% in feed	Chicken	8 weeks	*Enterobactericeace*	*Lactobacillus*	[63]
EGCG 0.6% in feed	Rat	4 weeks	*Clostridium*	None	[64]
Green tea extract 1.5 g/day in feed	Cow	5 weeks	*Clostridium*	*Bifidobacterium* and *Lactobacillus*	[65]
Green tea 1000 mL/day	Human	10 days	*Clostridium, Clostridium perfringens*	*Bifidobacterium*	[66]
Sunphenon (green tea catechin) 1.2 g/day	Human	4 weeks	None	*Bifidobacterium*	[67]

^1^ Reduced caffeine.

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
