# Peer review of "Gut/Oral Bacteria Variability May Explain the High Efficacy of Green Tea in Rodent Tumor Inhibition and Its Absence in Humans"

_molecules, 2020, doi:10.3390/molecules25204753_

Round 1

Reviewer 1 Report

EVALUATION

Ref. Molecules-951941

Type of manuscript: Review

Title: Gut/Oral Bacteria Variability May Explain the High Efficacy of Green

Tea in Rodent Tumor Inhibition and its Absence in Humans

Authors: Guy Adami*, Christy Tangney, Joel Schwartz, Kim Chi Dang

Submitted to section: Natural Products Chemistry

Plant Polyphenols and Gut Health

In this review, the authors seem to propose the hypothesis that variability in microbiota composition may be behind the lack of effect of green tea polyphenols against cancer in humans as compared with rodents where the antitumor properties of these compounds appear to be more consistent. The concept of interindividual variability in humans in response to dietary bioactive compounds and the factors influencing this response, i.e. genetic factors. microbiota composition and ADME among others, has been widely and thoroughly analysed and reported in several publications by researchers working in the area and reunited in a very successful COST European action (POSITIVe FA https://www6.inrae.fr/cost-positive/Dissemination/Publications-Outcomes) and thus, the main outcome of the article, the model summarized in Figure 1 is not entirely novel. However, the idea of collecting and comparing the results attained in the animal cancer models against human trials can be interesting and helpful in the search for potential reasons underlying the differences between the results attained in the two types of studies. After careful reading of the manuscript, I think that it truly needs a thorough revision before it may be considered for publication. Next, I exposed some of the issues that, in my opinion, need to be improved:

  • There is a general lack of focus in the article. In the text, the authors mix results from many different types of cancers (e.g. oral, oesophagus, colon, liver, breast, prostate…) but the general idea appears to be that they are mostly referring to the oral cavity tumours (right?). This is not clear. They should focus only in one group of cancer diseases, for example, gastrointestinal tract (oral, oesophagus, colon) or even only, oral cancers, otherwise the subject becomes too complex and confusing.
  • Along the same lines, the authors should also focus on gut or oral microbiota, depending on the cancer diseases objective of the study. I sort of understand that the main idea goes around oral microbiota and oral cancer, so, please focus on a particular subject, otherwise the whole exercise results a bit too vague and looses power and meaning.
  • Table 1. I think is of no use the way it is presented. First of all, the authors should include the other 5 studies (ref: 6, 16-19) where it appears that the GT polyphenols are not successful in rodents which takes us to 5 studies in favour and 5 studies against. In my view, this is not consistent evidence (!!!). On the other hand, this table should better include essential information regarding the aim of the review: i) the tumour inhibition results of each study. How good are the GT polyphenols in animals? 100% inhibition? 30% inhibition?, ii) the tumour model (chemically induced, human cancer cells injected in the animal, etc,) and iii) the human equivalent dose (very important). All these aspects are essential to understand the potential similarities and (or) differences between the studies and thus, the potential conclusion, or not, of the effects of these compounds against oral cancer (or gut cancers).
  • Table 2: the same comment. I would specify the doses so that they can be better compared with the animal models as well as the final results in tumour inhibition and variability in the results if this was indicated in the respective studies.
  • Metabolism: this is indeed a critical factor that needs also to be clarified in order to contribute to the understanding of the effects on GT polyphenols against cancer or any other health effect attributed to these compounds. Although there is a section dedicated to it in the review, once again because of the lack of focus, I don’t know what of the described metabolites are of interest in the case of oral cancer and why. Indeed, drinking tea is a very rapid process that does not allow the compounds to remain in the mouth in contact with oral epithelium for very long (seconds?) and thus, it is plausible that the potential anticancer effects may be mediated via circulating metabolites. This all needs to be better explained. The text is again confusing and even erroneous. For example, the authors only talk about bacterial enzymes involved in the metabolism but they mention phase I and phase II metabolites (i.e. glucuronides, methylated, etc.) (these metabolites are produced by intestinal and liver enzymes!!!). The whole issue of GT metabolism must be clarified and focused on the type of cancer investigated as well as on the related microbiota.  
  • Last, but not least, I would like to comment on the genetic part of the review. I sort of guess that the authors try to infer that gene expression changes associated to cancer prevention in rodents’ oral epithelium are more consistent than in humans because of the general homogeneity in the animals (diet, housing, genetic background, microbiota) than in the humans. In my opinion, this part is poorly written and a bit messy. The authors again mix genes, microRNAs and even proteins and it is all vague and confusing. Perhaps, an additional table with a collection of those studies reporting true gene changes in response to GT polyphenols in rodents and human studies of oral cancer prevention could help to clarify a bit better this point. How consistent and significant are the reported changes?

Overall, as I have already stated, this review needs a substantial revision of articles, presentation of data, focus and clarification of the main idea proposed by the authors.

Author Response

We thank the reviewer for the insightful and accurate comments. We have thought a lot about them and made many of the changes that were suggested but for reasons below we could not make all of them. A short description of the changes and the explanations are below.

Title: Gut/Oral Bacteria Variability May Explain the High Efficacy of Green

Tea in Rodent Tumor Inhibition and its Absence in Humans

Authors: Guy Adami*, Christy Tangney, Joel Schwartz, Kim Chi Dang

Submitted to section: Natural Products Chemistry

Plant Polyphenols and Gut Health

In this review, the authors seem to propose the hypothesis that variability in microbiota composition may be behind the lack of effect of green tea polyphenols against cancer in humans as compared with rodents where the antitumor properties of these compounds appear to be more consistent. The concept of interindividual variability in humans in response to dietary bioactive compounds and the factors influencing this response, i.e. genetic factors. microbiota composition and ADME among others, has been widely and thoroughly analysed and reported in several publications by researchers working in the area and reunited in a very successful COST European action (POSITIVe FA https://www6.inrae.fr/cost-positive/Dissemination/Publications-Outcomes) and thus, the main outcome of the article, the model summarized in Figure 1 is not entirely novel. However, the idea of collecting and comparing the results attained in the animal cancer models against human trials can be interesting and helpful in the search for potential reasons underlying the differences between the results attained in the two types of studies. After careful reading of the manuscript, I think that it truly needs a thorough revision before it may be considered for publication. Next, I exposed some of the issues that, in my opinion, need to be improved:

  • There is a general lack of focus in the article. In the text, the authors mix results from many different types of cancers (e.g. oral, oesophagus, colon, liver, breast, prostate…) but the general idea appears to be that they are mostly referring to the oral cavity tumours (right?). This is not clear. They should focus only in one group of cancer diseases, for example, gastrointestinal tract (oral, oesophagus, colon) or even only, oral cancers, otherwise the subject becomes too complex and confusing.

The reviewer is correct it would be a more straightforward review if the entire focus was on one cancer type such as OSCC. Other cancers were brought in because the general poor inhibition of carcinogenesis by GT in human versus rodent is true for many cancer types, and there are just not enough in vivo studies done to make a case about any specific cancer and inhibition by green tea, while also discussing digestive tract bacteria, and gene expression changes in human and rodent tissue under conditions for cancer formation. After all the only tissues that can easily and nonsurgically be sampled for RNA expression in cancer prevention study  is oral epithelium and blood.

Much of the ongoing  investigation of interindividual differences in polyphenol responses is not on cancer but other chronic diseases, but we have added a reference from the POSITVe site, (Espin et al. 2017) and the link itself,  to address this omission and to recognize that work which we agree is of great importance.

Addressed in Line 180-182

  • Along the same lines, the authors should also focus on gut or oral microbiota, depending on the cancer diseases objective of the study. I sort of understand that the main idea goes around oral microbiota and oral cancer, so, please focus on a particular subject, otherwise the whole exercise results a bit too vague and loses power and meaning.

There is ample evidence for microbial metabolism of GT metabolites in the gut but studies of gut epithelium gene expression are seldom if ever done in humans. We have now made it more obvious that polyphenol metabolites from the gut go to the oral epithelium via the blood, though it is not known how important that is. Besides Wang’s study in 1999 there is little known about oral cavity processing of GT metabolites. Though GT has large effects on makeup of saliva microbiome further supporting the possibility there are oral microbes that can metabolize it. But we have shortened the speculation on the contribution of oral bacteria to catechin metabolism.

Addressed in Lines 193-198

  • Table 1. I think is of no use the way it is presented. First of all, the authors should include the other 5 studies (ref: 6, 16-19) where it appears that the GT polyphenols are not successful in rodents which takes us to 5 studies in favour and 5 studies against. In my view, this is not consistent evidence (!!!). On the other hand, this table should better include essential information regarding the aim of the review: i) the tumour inhibition results of each study. How good are the GT polyphenols in animals? 100% inhibition? 30% inhibition?, ii) the tumour model (chemically induced, human cancer cells injected in the animal, etc,) and iii) the human equivalent dose (very important). All these aspects are essential to understand the potential similarities and (or) differences between the studies and thus, the potential conclusion, or not, of the effects of these compounds against oral cancer (or gut cancers).

As suggested by the reviewer we have made Table 1 more informative and have tried to make it less confusing. There are only 5 rodent studies in the last 25 years using GT or GT polyphenols to inhibit OSCC and they are all now presented in table 1.  We tried to limit studies to those done in the last 25 years, when there was more attention to how much GT polyphenol the animals actually consumed. The one study that used syngeneic mouse cell lines in mice  is now marked as such. And we have now marked whether GT and carcinogen were present concurrently or not. The purpose of Table 1 is in part to demonstrate that rodent studies use variable amounts of GT extract or polyphenols within the same species. And that it is typically delivered orally in liquid form though not always. We have added one study and have changed time of exposure to refer only to time of exposure to green tea catechins and have added assays of cancer inhibition as requested.

See new Table 1.

  • Table 2: the same comment. I would specify the doses so that they can be better compared with the animal models as well as the final results in tumour inhibition and variability in the results if this was indicated in the respective studies.

The doses are supplied in mg/ KG subject mass. I think that these are fairly comparable. If the reviewer wants the doses to be supplied in mg per surface area that can be done also.

There are only two studies on GT polyphenol consumption and effects on oral cancer progression that have been reported and the outcomes are better highlighted in the text.  The Yu et al. study only looks at GT polyphenol toxicity and does not examine the effect on oral cancer formation. We changed line 68 in the study to be clearer about the results of these two OSCC studies and the toxicity study in Table 2.

  • Metabolism: this is indeed a critical factor that needs also to be clarified in order to contribute to the understanding of the effects on GT polyphenols against cancer or any other health effect attributed to these compounds. Although there is a section dedicated to it in the review, once again because of the lack of focus, I don’t know what of the described metabolites are of interest in the case of oral cancer and why. Indeed, drinking tea is a very rapid process that does not allow the compounds to remain in the mouth in contact with oral epithelium for very long (seconds?) and thus, it is plausible that the potential anticancer effects may be mediated via circulating metabolites. This all needs to be better explained. The text is again confusing and even erroneous. For example, the authors only talk about bacterial enzymes involved in the metabolism but they mention phase I and phase II metabolites (i.e. glucuronides, methylated, etc.) (these metabolites are produced by intestinal and liver enzymes!!!). The whole issue of GT metabolism must be clarified and focused on the type of cancer investigated as well as on the related microbiota.  

First of all we thank the reviewer for pointing out our error of not differentiating the bacterial GT catechin metabolism versus the different reactions that occur chiefly in the liver by host enzymes. We have corrected this. We have added a figure to show some of the initial metabolites formed during the bacterial based breakdown of one GT catechin, EC.

The tea is in the mouth long enough to affect bacteria so there are likely to be direct effects on oral epithelium. Though absorption of GT polyphenols and their metabolites in the gut, which travel by blood to the oral epithelium could certainly occur. We shorten this part on mouth catechin metabolism and discuss blood transport of tea metabolites.

Corrected in Lines 175-182  and 196-199.

Little  is known about the direct effects of GT polyphenol and their derivatives on epithelial cells in vivo like EGCG and it has been studied for over 25 years. There is a good deal of research on the formation of  phenyl-γ-valerolactones after polyphenol consumption and the activity of these compounds in changing cell phenotype, though little of this work has been done on carcinogenesis in vitro and less in vivo.

  • Last, but not least, I would like to comment on the genetic part of the review. I sort of guess that the authors try to infer that gene expression changes associated to cancer prevention in rodents’ oral epithelium are more consistent than in humans because of the general homogeneity in the animals (diet, housing, genetic background, microbiota) than in the humans. In my opinion, this part is poorly written and a bit messy. The authors again mix genes, microRNAs and even proteins and it is all vague and confusing. Perhaps, an additional table with a collection of those studies reporting true gene changes in response to GT polyphenols in rodents and human studies of oral cancer prevention could help to clarify a bit better this point. How consistent and significant are the reported changes?

The vast majority of studies on GT polyphenols and gene expression are done in vitro with cell lines. There are only a few studies published on GT polyphenol effects on epithelial gene expression (or RNA levels)  in vivo in rodents and none on oral epithelium. The examples we highlight, two on lung cancer and one on liver cancer, were chosen because in those models of cancer inducing agent was present prior to GT polyphenols being introduced. Other examples where GT polyphenols are administered to the subjects and they are also challenged with carcinogen at the same time are listed but are not discussed at length. We added examples of changes in protein levels as it addresses the same point, do the GT polyphenols have reproducible effects on the levels of gene products found in epithelium in vivo?

Corrected in part in lines 107 -109.

Overall, as I have already stated, this review needs a substantial revision of articles, presentation of data, focus and clarification of the main idea proposed by the authors.

Reviewer 2 Report

Overall the manuscript is interesting and well-written. I have one suggestions, as for the metabolism of the green tea, which bacteria is responsible for it and which enzyme is responsible for it? These should be discussed in the manuscript and I suggest the authors add another figure to illustrate it. 

Author Response

We thank the reviewer for his positive comments on the manuscript. And especially for the suggestion to add a figure on catechin metabolism. We have added figure 4 on the products of epicatechin (EC) metabolism assigned to specific gut bacteria. These are products that occur early in the process of catechin metabolism and are related to the parent compound so are likely to be bioactive.. These metabolic products have been shown to have effects on cells in the few studies that have been done, though relevance to carcinogenesis is not known.

Round 2

Reviewer 1 Report

The paper may be accepted for publication since it has been revised by the authors following most of the suggestions and, the main ideas conveyed by the review are clearly exposed, i.e. in vivo animal results are more homogeneous than human trials and, microbiota and its metabolic contribution are likely to be partially responsible for the variability in the results in humans. These ideas, although not novel in the area of the beneficial effects of polyphenols, are newly and specifically applied to the effects of green tea polyphenols against cancer in this review.